# Trigeminal Neuralgia Treatment Outcomes Following Gamma Knife Stereotactic Radiosurgery

Abbas Jarrahi [1], Rebecca Cantrell [2], Cynthia Norris [2], Krishnan Dhandapani [1,*], John Barrett [2] and John Vender [1,*]

1   Department of Neurosurgery, Medical College of Georgia, Augusta University, Augusta, GA 30912, USA
2   Department of Radiation Oncology, Medical College of Georgia, Augusta University, Augusta, GA 30912, USA
*   Correspondence: kdhandapani@augusta.edu (K.D.); jvender@augusta.edu (J.V.);
    Tel.: +1(706)-721-8846 (K.D.); +1(706)-721-3071 (J.V.)

**Abstract:** Trigeminal neuralgia (TN) is a chronic pain condition causing lancinating pain in the distribution of one or more divisions of the trigeminal nerve. Gamma knife stereotactic radiosurgery (GKSRS) is a surgical option for TN refractory to medical therapy. To report our experience and to analyze the reasons for the variance in radiosurgery outcomes between patients in our diverse population, we conducted a retrospective analysis of a prospectively created database. The 178 patients completed a pain assessment questionnaire before surgery, and at 1 and 2 year follow-ups. We used the "Trigeminal Neuralgia Gamma Knife Outcome Scale" (TN GKOS) to report the response. At 1-year, 35.4% of patients had grade 1A outcome (pain-free and off all pain medications), 24.7% had grade 1B (pain-free on pain medications), 24.2% had grade 1C (some pain but improved with radiosurgery), 12.9% had grade 2 (same as before radiosurgery) and 2.8% had grade 3 (worse pain compared to before radiosurgery). At 2 years, 42.3% had grade 1A, 20.5% had grade 1B, 19.2% had grade 1C, 14.1% had grade 2 and 3.8% had grade 3 outcome. Remarkably, a statistically significant association was found between GKOS and age, racial background and obesity.

**Keywords:** trigeminal neuralgia; gamma knife; follow-up; clinical outcomes; age; obesity

## 1. Introduction

Trigeminal neuralgia (TN) is a chronic pain condition that causes lancinating pain in the distribution of one or more divisions of the trigeminal (5th cranial) nerve, which is sudden episodic and/or constant in nature. The first descriptions of TN date back to the 11th century, in The Canon (Laws of Medicine) book by the Persian physician and philosopher Avicenna [1]. More accurate descriptions came in the 18th century, first by Nicholas André who termed the condition tic douloureux (painful tic in French); and then by John Fothergill in London [2]. Facial pain can be due to a number of conditions including TN [3–6]. TN is classified into primary TN, which includes typical type 1 TN (predominantly paroxysmal pain) and atypical type 2 TN (predominantly constant pain); and secondary TN [7]. Although medical treatment is first-line management for most patients, surgical therapy is an option for patients with medication intolerance or TN refractory to medication. Surgical options include microvascular decompression (MVD), percutaneous ablative procedures on the Gasserian ganglion (mechanical balloon compression, chemical (glycerol) rhizolysis and radiofrequency thermocoagulation rhizotomy) and stereotactic radiosurgery (SRS), including gamma knife stereotactic radiosurgery (GKSRS), Linac (linear accelerator) SRS and CyberKnife SRS. GK uses high doses (70 to 90 Gy) of subcentimeter radiation beams focused on the trigeminal pontine root entry zone (REZ). The focused gamma radiations cause focal axonal degeneration and necrosis over time, thus decreasing pain signals [8]. This process takes time, and although the time to response varies between individuals, the average time interval from radiation to symptom improvement is approximately one month [9,10]. In a systematic review, after GKS the initial freedom from

pain without medication ranged from 28.6% to 100%, with a mean of 53.1% and a median of 52.1%. At 10 years following GKS, 30% to 45.3% of patients were pain free without medication [11]. There is no explanation for this wide spectrum of clinical results and why some patients respond better than others. We aim to report our experience in use of GKSRS for patients with TN and to analyze the reason for this change in efficacy of GK outcome in our diverse population.

## 2. Materials and Methods

*Patient population:* We conducted a retrospective analysis of a prospectively created database of TN patients who underwent GKSRS at Augusta University Gamma Knife Center. The database recorded patients from 2000 to 2020 with a total of 587 patients. Of that population, 178 patients were followed up at 1 year and responded to the assessment questionnaire. At 2 years, 78 patients were assessed. Additionally, our center began performing third GKSRS treatments in select patients, therefore this population of patients will accrue over time. We received a waiver of consent as a chart review from the institutional review board office at Augusta University (file number 08-03-229).

*Radiosurgical technique:* Radiosurgery was performed using Elekta Gamma Knife B Unit (from May 2000 to July 2011) and then using Leksell Gamma Knife Perfexion (from August 2011 till present). Leksell stereotactic head frame is applied and neuroimaging for treatment planning is obtained using magnetic resonance imaging (MRI) or computed tomography (CT) with cisternography, if MRI is contraindicated. In all patients, the trigeminal nerve was treated using a single 4 mm collimator to the 100% isodose line. We used a maximal radiation dose of 80 Gy in patients undergoing their first GKS, a dose of 70 Gy during second treatments (46 patients) and a dose of 60 Gy during third treatments (5 patients). In retreatment cases different target locations were selected according to standard guidelines.

*Patient follow-up:* Enrolled patients completed a pre-GK pain assessment questionnaire before undergoing GKS, where they delineated pain site, onset, character, radiation, frequency & timing, alleviating & aggravating factors, and severity. With respect to pain severity evaluation, a numeric rating scale of 0 to 10 was utilized ('0' representing no pain and '10' representing the worst imaginable pain). Patients used the scale to rate the maximum and average level of their pain. Additionally, they stated the impact of pain on their quality of life and presence of any chronic comorbid conditions. Follow-up questionnaires were sent to patients by mail or filled verbally via phone calls by a trained nurse at 12 and 24 months. We developed an outcome scale to assess patients' status following GKS. The scale named "Trigeminal Neuralgia Gamma Knife Outcome Scale" (TN GKOS) (Table 1), is a modified version of the Barrow Neurological Institute (BNI) pain intensity scoring criteria [12], and defines the response to surgery in 5 categories. A score of 1 represents a good response to surgery and is further subcategorized into 1A, where patients have no pain without taking medications; 1B, where patients are pain-free with medications; and 1C where they have some pain, but they had pain improvement with surgery. Grades 2 and 3 represent same pain compared to before GKSRS and worse pain compared to before GKSRS, respectively. Furthermore, we assessed frequency and severity of pain, presence of postoperative facial numbness or tingling and pain recurrence.

*Statistical analyses:* Statistical analyses were performed using IBM Statistical Package for the Social Sciences (SPSS) software version 27 and GraphPad Prism 9 software. We conducted descriptive statistics to elucidate the mean, median, standard deviation, minimum, maximum and range values wherever appropriate. We analyzed the association between GK outcome and age, gender, race, pain side, pain division, atypical pain and comorbidities (diabetes mellitus, hypertension, hyperlipidemia, obesity, hypothyroidism, multiple sclerosis, meningioma, stroke, dementia, neurological dysfunction, seizures, familial tremor, psychiatric disorders and temporomandibular joint dysfunction). Patients with a body mass index (BMI) greater than or equal to 30, were considered obese. We used Pearson's Chi-square test to assess the association between categorical variables as appropriate. We

used ordinal logistic regression analysis to evaluate the relationship between continuous and categorical variables. Furthermore, multigroup comparisons of continuous variables were made using one-way ANOVA with Tukey's multiple comparisons test. Statistical significance was set at a *p*-value $\leq 0.05$.

**Table 1.** Trigeminal Neuralgia Gamma Knife Outcome Scale.

| Trigeminal Neuralgia Gamma Knife Outcome Scale (TN GKOS) | |
|---|---|
| **Score** | **Description** |
| 1A | Pain-free & off all pain medications |
| 1B | Pain-free on pain medications |
| 1C | Some pain, improved with GKSRS |
| 2 | Same as before GKSRS |
| 3 | Worse pain compared to before GKSRS |

## 3. Results

### 3.1. Clinical Outcomes

The mean age of the patients before the procedure was 67.4 years, median was 69, mode was 72 and range was 64 (35–99) years. Forty-eight patients (27%) were adults (18–60 years), and 130 patients (73%) were geriatric (>60 years). In our cohort, we had 62 (34.8%) male and 116 (65.2%) female patients. In terms of racial distribution, 21 (11.8%) were African American, 153 (86%) were Caucasian and 4 (2.2%) were Hispanic (Table 2). Additionally, 80 patients (50%) presented with right-sided pain, 77 (48.1%) had left-sided pain and 3 (1.9%) had bilateral pain. Furthermore, 7 (4.4%) patients experienced atypical pain. The prevalence of pain in the distribution of the different divisions of the trigeminal nerve is illustrated in Figure 1. We further demonstrated the time interval from first onset of TN to GKSRS (Figure 2), with a mean of 7.8 years, standard deviation of 7.2, median of 6, mode of less than or equal to 1 year, maximum of 42 years and range of 41 years. The results show that most patients (22) underwent GKSRS in less than or equal to 1 year since first onset of TN pain, with the number declining as the time interval expands.

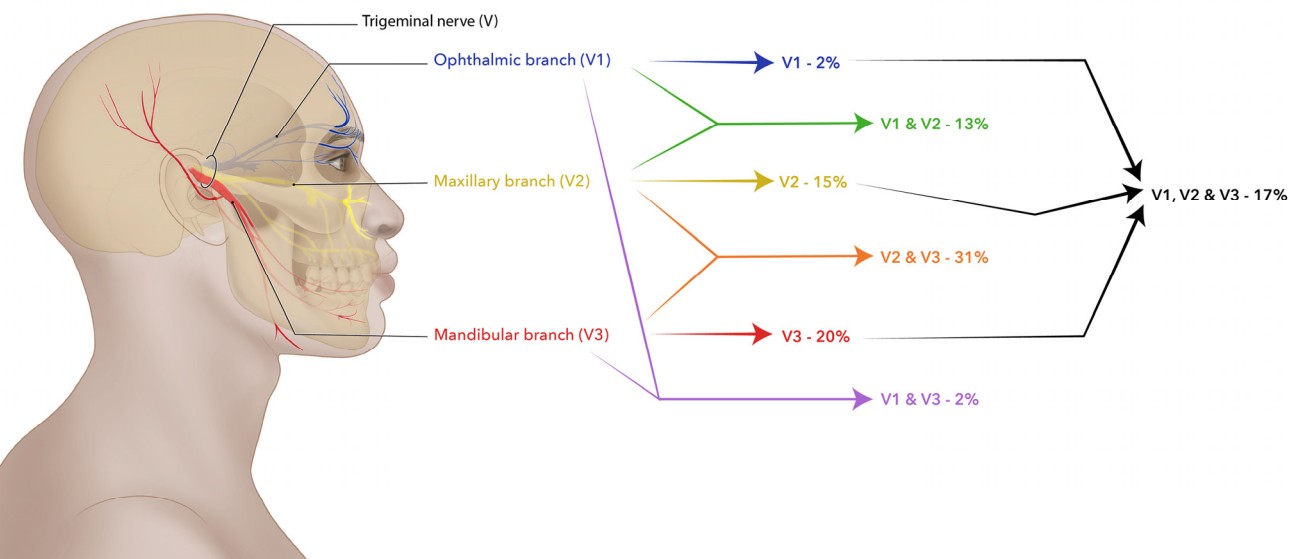

**Figure 1.** The prevalence of pain in the distribution of the different divisions of the trigeminal nerve.

**Table 2.** Trigeminal neuralgia patient demographics and presentation.

| Variable | Category | Number Percentage % | TN GKOS at 1 Year | | | | | *p*-Value | TN GKOS at 2 Years | | | | | *p*-Value |
|---|---|---|---|---|---|---|---|---|---|---|---|---|---|---|
| | | | 1A | 1B | 1C | 2 | 3 | | 1A | 1B | 1C | 2 | 3 | |
| Age | Adult (18–60) | 48 / 27 | 10 / 20.8 | 8 / 16.7 | 17 / 35.4 | 10 / 20.8 | 3 / 6.3 | 0.005 | 5 / 25.0 | 4 / 20.0 | 6 / 30.0 | 3 / 15.0 | 2 / 10.0 | 0.185 |
| | Geriatric (>60) | 130 / 73 | 53 / 40.8 | 36 / 27.7 | 26 / 20.0 | 13 / 10.0 | 2 / 1.5 | | 28 / 48.3 | 12 / 20.7 | 9 / 15.5 | 8 / 13.8 | 1 / 1.7 | |
| Gender | Male | 62 / 34.8 | 26 / 41.9 | 12 / 19.4 | 12 / 19.4 | 11 / 17.7 | 1 / 1.6 | 0.252 | 12 / 48.0 | 6 / 24.0 | 3 / 12.0 | 3 / 12.0 | 1 / 4.0 | 0.806 |
| | Female | 116 / 65.2 | 37 / 31.9 | 32 / 27.6 | 31 / 26.7 | 12 / 10.3 | 4 / 3.4 | | 21 / 39.6 | 10 / 18.9 | 12 / 22.6 | 8 / 15.1 | 2 / 3.8 | |
| Race | African American | 21 / 11.8 | 4 / 19.0 | 2 / 9.5 | 9 / 42.9 | 6 / 28.6 | 0 / 0.0 | 0.013 | 3 / 30.0 | 0 / 0.0 | 5 / 50.0 | 2 / 20.0 | 0 / 0.0 | 0.180 |
| | Caucasian | 153 / 86.0 | 58 / 37.9 | 42 / 27.5 | 31 / 20.3 | 17 / 11.1 | 5 / 3.3 | | 29 / 43.9 | 16 / 24.2 | 9 / 13.6 | 9 / 13.6 | 3 / 4.5 | |
| | Hispanic | 4 / 2.2 | 1 / 25.0 | 0 / 0.0 | 3 / 75.0 | 0 / 0.0 | 0 / 0.0 | | 1 / 50.0 | 0 / 0.0 | 1 / 50.0 | 0 / 0.0 | 0 / 0.0 | |
| Pain side | Right | 80 / 50.0 | 29 / 36.3 | 20 / 25.0 | 16 / 20.0 | 13 / 16.3 | 2 / 2.5 | 0.038 | 18 / 54.5 | 5 / 15.2 | 4 / 12.1 | 6 / 18.2 | 0 / 0.0 | 0.014 |
| | Left | 77 / 48.1 | 26 / 33.8 | 20 / 26.0 | 23 / 29.9 | 7 / 9.1 | 1 / 1.3 | | 13 / 39.4 | 8 / 24.2 | 9 / 27.3 | 2 / 6.1 | 1 / 3.0 | |
| | Bilateral | 3 / 1.9 | 1 / 33.3 | 0 / 0.0 | 1 / 33.3 | 0 / 0.0 | 1 / 33.3 | | 0 / 0.0 | 0 / 0.0 | 1 / 33.3 | 1 / 33.3 | 1 / 33.3 | |
| Pain division | V1 | 3 / 1.9 | 0 / 0.0 | 2 / 66.7 | 0 / 0.0 | 1 / 33.3 | 0 / 0.0 | 0.751 | 0 / 0.0 | 0 / 0.0 | 0 / 0.0 | 2 / 100 | 0 / 0.0 | 0.161 |
| | V2 | 24 / 15.2 | 7 / 29.2 | 6 / 25.0 | 9 / 37.5 | 2 / 8.3 | 0 / 0.0 | | 3 / 30.0 | 3 / 30.0 | 4 / 40.0 | 0 / 0.0 | 0 / 0.0 | |
| | V3 | 32 / 20.3 | 11 / 34.4 | 9 / 28.1 | 8 / 25.0 | 4 / 12.5 | 0 / 0.0 | | 9 / 69.2 | 1 / 7.7 | 2 / 15.4 | 1 / 7.7 | 0 / 0.0 | |
| | V1, V2 | 21 / 13.3 | 9 / 42.9 | 2 / 9.5 | 4 / 19.0 | 5 / 23.8 | 1 / 4.8 | | 6 / 54.5 | 1 / 9.1 | 2 / 18.2 | 1 / 9.1 | 1 / 9.1 | |
| | V1, V3 | 3 / 1.9 | 2 / 66.7 | 0 / 0.0 | 1 / 33.3 | 0 / 0.0 | 0 / 0.0 | | 2 / 66.7 | 0 / 0.0 | 1 / 33.3 | 0 / 0.0 | 0 / 0.0 | |
| | V2, V3 | 49 / 31.0 | 18 / 36.7 | 14 / 28.6 | 12 / 24.5 | 3 / 6.1 | 2 / 4.1 | | 8 / 36.4 | 6 / 27.3 | 4 / 18.2 | 4 / 18.2 | 0 / 0.0 | |
| | V1, V2, V3 | 26 / 16.5 | 8 / 30.8 | 7 / 26.9 | 6 / 23.1 | 4 / 15.4 | 1 / 3.8 | | 3 / 37.5 | 2 / 25.0 | 1 / 12.5 | 1 / 12.5 | 1 / 12.5 | |
| Atypical pain | Yes | 7 / 4.4 | 0 / 0.0 | 1 / 14.3 | 3 / 42.9 | 3 / 42.9 | 0 / 0.0 | 0.052 | 0 / 0.0 | 0 / 0.0 | 0 / 0.0 | 2 / 100 | 0 / 0.0 | 0.008 |
| | No | 153 / 95.6 | 56 / 36.6 | 39 / 25.5 | 37 / 24.2 | 17 / 11.1 | 4 / 2.6 | | 31 / 46.3 | 13 / 19.4 | 14 / 20.9 | 7 / 10.4 | 2 / 3.0 | |

A comprehensive description of pain character, along with aggravating and alleviating factors and impact on quality of life before GKSRS was noted (Table 3). Eighteen patients did not respond to the pre-GK assessment questionnaire relating to pain characterization, but they were included since they responded to the other questions. In these cases, we calculated the valid percentages excluding the missing values. The most common aggravating factors were eating, talking, touch and brushing teeth. Rest was the second most frequently reported alleviating factor after medication use. Of note, cold made the pain worse in 60% (96) but also mitigated the pain in 5% (8) of patients, whereas a similar number of patients reported that heat made their pain better 23.1% (37) or worse 23.8% (38). Only 6.3% of patients reported that their TN pain did not change their quality of life. The most stated impact was the reduction in physical activity (48.8%) followed by sleep changes (43.8%), appetite reduction (30.6%), altered relationships (26.9%) and emotional effects (19.4%).

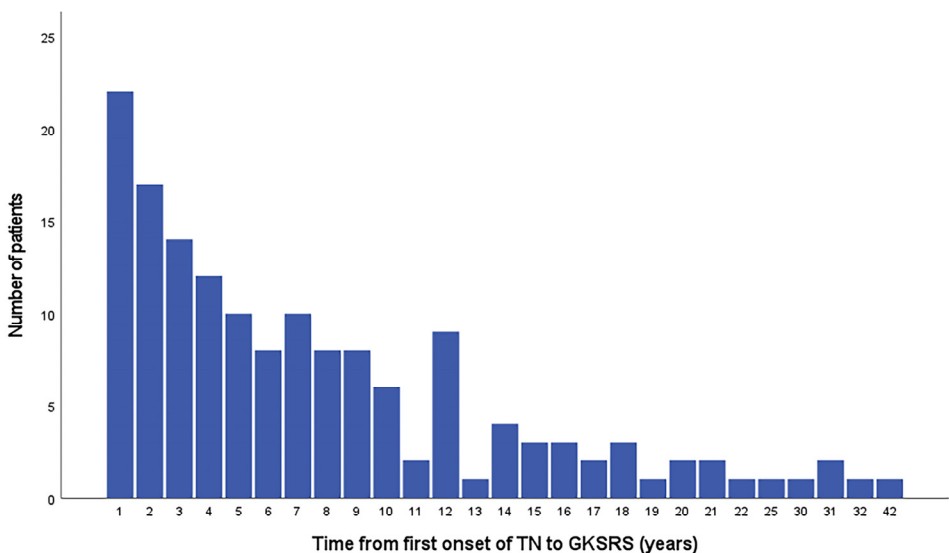

**Figure 2.** Time interval from the first onset of trigeminal neuralgia to GKSRS.

**Table 3.** Trigeminal Neuralgia Pain Characteristics.

| Category | Variable | Number (Percentage %) | Category | Variable | Number (Percentage %) |
|---|---|---|---|---|---|
| Pain character | | | Aggravating factors | | |
| | Stabbing pain | 98 (61.3) | | Activity | 52 (32.5) |
| | Electrical shock pain | 134 (83.8) | | Eating | 135 (84.4) |
| | Sharp pain | 136 (85.0) | | Heat | 38 (23.8) |
| | Dull pain | 49 (30.6) | | Positioning | 85 (53.1) |
| | Aching pain | 53 (33.1) | | Talking | 128 (80.0) |
| | Tender pain | 79 (49.4) | | Cold | 96 (60.0) |
| | Pressure pain | 48 (30.0) | | Coughing/Deep breaths | 53 (33.1) |
| | Throbbing pain | 88 (55.0) | | Touch | 119 (74.4) |
| | Cramping pain | 18 (11.3) | | Brushing teeth | 116 (72.5) |
| | Burning pain | 69 (43.1) | | Brushing hair | 50 (31.3) |
| | Pulling pain | 22 (13.8) | | Shaving | 37 (23.1) |
| Changes in quality of life | | | | Putting makeup | 54 (33.8) |
| | None | 10 (6.3) | Alleviating factors | | |
| | Sleep changes | 70 (43.8) | | Rest | 40 (25.0) |
| | Reduced appetite | 49 (30.6) | | Medication | 129 (80.6) |
| | Reduced physical activity | 78 (48.8) | | Heat | 37 (23.1) |
| | Emotional | 31 (19.4) | | Cold | 8 (5.0) |
| | Altered relationships | 43 (26.9) | | | |

Most patients had a good response (grade 1) at both 1 year (150, 84.3%) and 2 years (64, 82.1%). At 1 year, 35.4% of patients had grade 1A, 24.7% had grade 1B, 24.2% had grade 1C, 12.9% had grade 2 and 2.8% had grade 3 outcome. At the 2 years, 42.3% of patients had grade 1A, 20.5% had grade 1B, 19.2% had grade 1C, 14.1% had grade 2 and 3.8% had grade 3 outcome (Figure 3A,B). Additionally, ordinal logistic regression analysis showed that

time interval from first onset of TN to GKS was not associated with outcomes. Furthermore, we comprehensively illustrate the severity and frequency of pain before GKSRS and at the 1-year and 2 years follow-up assessments (Figures 4A–H and 5A–C). Using one-way ANOVA, we demonstrated that there was a statistically significant decline in the maximum and average pain severity scores at both assessments (Figure 4A–H). Along the same lines, a remarkable attenuation in pain frequency was observed on follow-up compared to pre-procedure assessment (Figure 5A–C). Nonetheless, no significant variation in pain severity was detected between the first and second assessment.

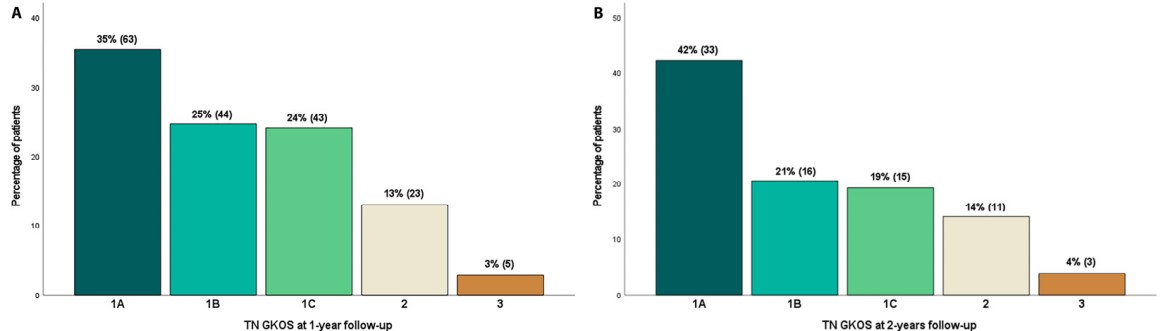

**Figure 3.** TN GKOS grades at 1 year follow-up (**A**), and 2 years follow-up (**B**).

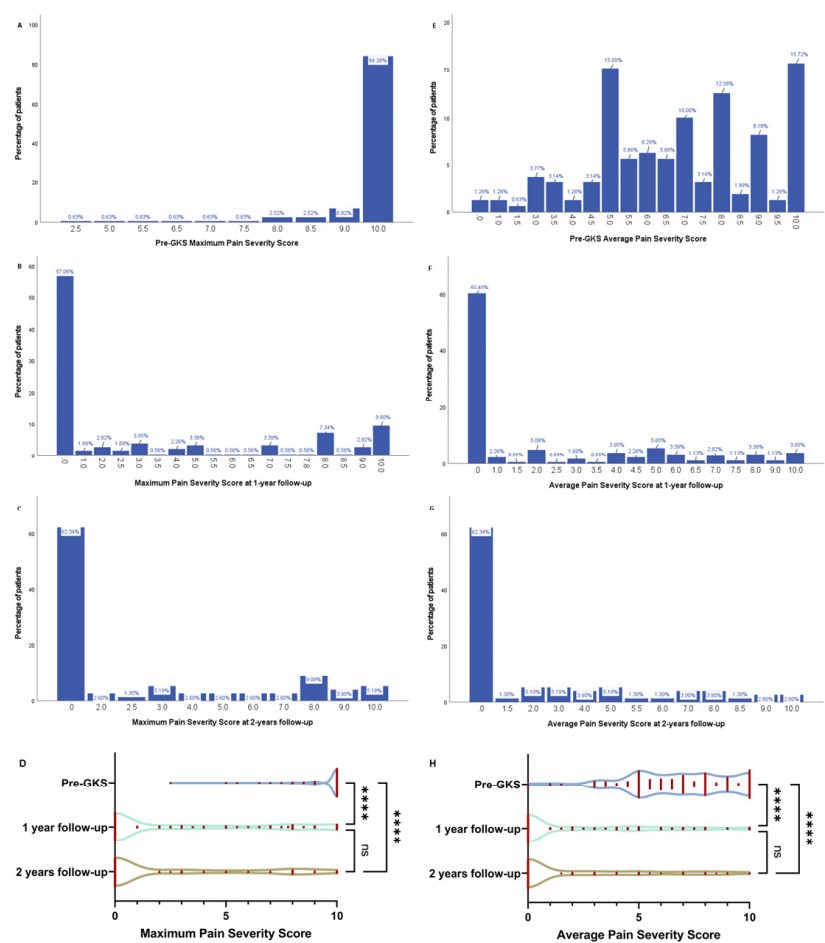

**Figure 4.** Pain severity before GKSRS, and at the 1-year and 2 years follow-up assessments, displaying the maximum (**A–D**) and average pain severity scores (**E–H**). ns: not significant, **** $p < 0.0001$.

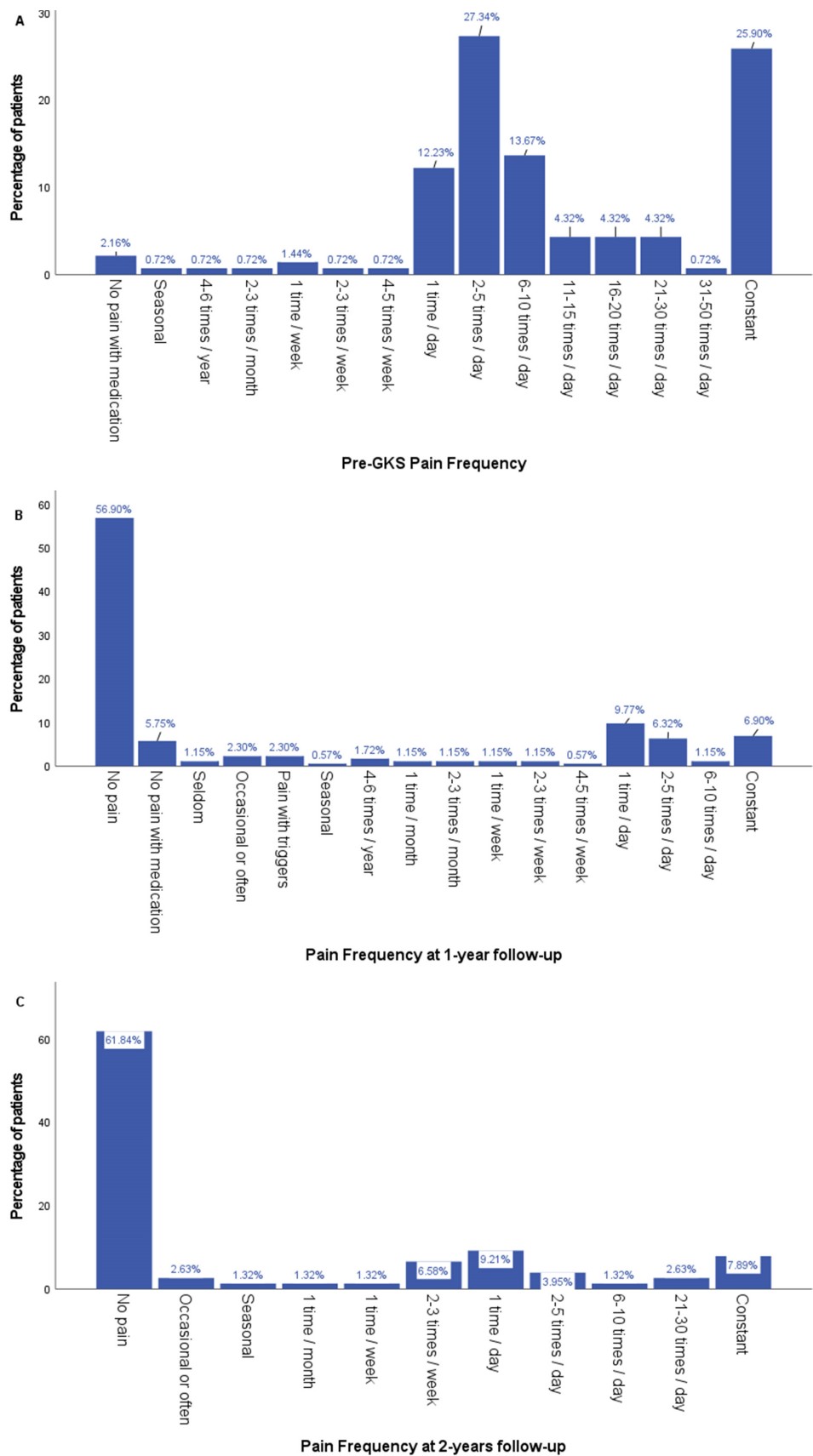

**Figure 5.** Pain frequency before GKSRS (**A**), and at the 1-year (**B**) and 2 years (**C**) follow-up assessments.



*3.2. Factors Associated with Good GKRSR Outcomes*

The association between several factors mentioned in the methods section and GKOS was assessed and displayed in Table 4.

**Table 4.** Trigeminal Neuralgia Pain Characteristics.

| Variable | | Number Percentage % | TN GKOS at 1 Year | | | | | *p*-Value | TN GKOS at 2 Years | | | | | *p*-Value |
|---|---|---|---|---|---|---|---|---|---|---|---|---|---|---|
| | | | 1A | 1B | 1C | 2 | 3 | | 1A | 1B | 1C | 2 | 3 | |
| Diabetes mellitus | Yes | 25 14.0 | 9 36.0 | 5 20.0 | 8 32.0 | 3 12.0 | 0 0.0 | 0.765 | 5 38.5 | 3 23.1 | 3 23.1 | 2 15.4 | 0 0.0 | 0.933 |
| | No | 153 86 | 54 35.3 | 39 25.5 | 35 22.9 | 20 13.1 | 5 3.3 | | 28 43.1 | 13 20.0 | 12 18.5 | 9 13.8 | 3 4.6 | |
| Hypertension | Yes | 85 47.8 | 27 31.8 | 23 27.1 | 22 25.9 | 12 14.1 | 1 1.2 | 0.576 | 15 41.7 | 9 25.0 | 7 19.4 | 4 11.1 | 1 2.8 | 0.864 |
| | No | 93 52.2 | 36 38.7 | 21 22.6 | 21 22.6 | 11 11.8 | 4 4.3 | | 18 42.9 | 7 16.7 | 8 19.0 | 7 16.7 | 2 4.8 | |
| Hyperlipidemia | Yes | 32 18.0 | 13 40.6 | 9 28.1 | 7 21.9 | 2 6.3 | 1 3.1 | 0.749 | 8 50.0 | 3 18.8 | 3 18.8 | 2 12.5 | 0 0.0 | 0.890 |
| | No | 146 82.0 | 50 34.2 | 35 24.0 | 36 24.7 | 21 14.4 | 4 2.7 | | 25 40.3 | 13 21.0 | 12 19.4 | 9 14.5 | 3 4.8 | |
| Obesity | Yes | 5 2.8 | 0 0.0 | 1 20.0 | 3 60.0 | 0 0.0 | 1 20.0 | 0.030 | 0 0.0 | 0 0.0 | 0 0.0 | 1 50.0 | 1 50.0 | 0.005 |
| | No | 173 97.2 | 63 36.4 | 43 24.9 | 40 23.1 | 23 13.3 | 4 2.3 | | 33 43.4 | 16 21.1 | 15 19.7 | 10 13.2 | 2 2.6 | |
| Hypothyroidism | Yes | 24 13.5 | 7 29.2 | 5 20.8 | 7 29.2 | 5 20.8 | 0 0.0 | 0.577 | 6 75.0 | 1 12.5 | 0 0.0 | 1 12.5 | 0 0.0 | 0.330 |
| | No | 154 86.5 | 56 36.4 | 39 25.3 | 36 23.4 | 18 11.7 | 5 3.2 | | 27 38.6 | 15 21.4 | 15 21.4 | 10 14.3 | 3 4.3 | |
| Multiple sclerosis | Yes | 5 2.8 | 1 20.0 | 2 40.0 | 1 20.0 | 1 20.0 | 0 0.0 | 0.876 | 1 100 | 0 0.0 | 0 0.0 | 0 0.0 | 0 0.0 | 0.847 |
| | No | 173 97.2 | 62 35.8 | 42 24.3 | 42 24.3 | 22 12.7 | 5 2.9 | | 32 41.6 | 16 20.8 | 15 19.5 | 11 14.3 | 3 3.9 | |
| Meningioma | Yes | 2 1.1 | 1 50.0 | 0 0.0 | 0 0.0 | 1 50.0 | 0 0.0 | 0.506 | 1 100 | 0 0.0 | 0 0.0 | 0 0.0 | 0 0.0 | 0.847 |
| | No | 176 98.9 | 62 35.2 | 44 25.0 | 43 24.4 | 22 12.5 | 5 2.8 | | 32 41.6 | 16 20.8 | 15 19.5 | 11 14.3 | 3 3.9 | |
| Neurological dysfunction | Yes | 2 1.1 | 0 0.0 | 2 100 | 0 0.0 | 0 0.0 | 0 0.0 | 0.188 | 1 100 | 0 0.0 | 0 0.0 | 0 0.0 | 0 0.0 | 0.847 |
| | No | 176 98.9 | 63 35.8 | 42 23.9 | 43 24.4 | 23 13.1 | 5 2.8 | | 32 41.6 | 16 20.8 | 15 19.5 | 11 14.3 | 3 3.9 | |
| Stroke | Yes | 12 6.7 | 6 50.0 | 4 33.3 | 1 8.3 | 1 8.3 | 0 0.0 | 0.547 | 4 66.7 | 0 0.0 | 1 16.7 | 1 16.7 | 0 0.0 | 0.636 |
| | No | 166 93.3 | 57 34.3 | 40 24.1 | 42 25.3 | 22 13.3 | 5 3.0 | | 29 40.3 | 16 22.2 | 14 19.4 | 10 13.9 | 3 4.2 | |
| Dementia | Yes | 2 1.1 | 0 0.0 | 1 50.0 | 1 50.0 | 0 0.0 | 0 0.0 | 0.714 | | | | | | |
| | No | 176 98.9 | 63 35.8 | 43 24.4 | 42 23.9 | 23 13.1 | 5 2.8 | | | | | | | |
| Seizures | Yes | 1 0.6 | 0 0.0 | 0 0.0 | 0 0.0 | 1 100 | 0 0.0 | 0.148 | | | | | | |
| | No | 177 99.4 | 63 35.6 | 44 24.9 | 43 24.3 | 22 12.4 | 5 2.8 | | | | | | | |
| Familial tremor | Yes | 1 0.6 | 0 0.0 | 0 0.0 | 0 0.0 | 1 100 | 0 0.0 | 0.148 | 0 0.0 | 0 0.0 | 1 100 | 0 0.0 | 0 0.0 | 0.373 |
| | No | 177 99.4 | 63 35.6 | 44 24.9 | 43 24.3 | 22 12.4 | 5 2.8 | | 33 42.9 | 16 20.8 | 14 18.2 | 11 14.3 | 3 3.9 | |
| Psychiatric disorders (depression, bipolar, anxiety, sleep disorders) | Yes | 18 10.1 | 6 33.3 | 4 22.2 | 5 27.8 | 1 5.6 | 2 11.1 | 0.204 | 3 50.0 | 1 16.7 | 1 16.7 | 1 16.7 | 0 0.0 | 0.979 |
| | No | 160 89.9 | 57 35.6 | 40 25.0 | 38 23.8 | 22 13.8 | 3 1.9 | | 30 41.7 | 15 20.8 | 14 19.4 | 10 13.9 | 3 4.2 | |
| Temporomandibular joint dysfunction | Yes | 2 1.1 | 1 50.0 | 0 0.0 | 1 50.0 | 0 0.0 | 0 0.0 | 0.827 | 0 0.0 | 1 50.0 | 1 50.0 | 0 0.0 | 0 0.0 | 0.538 |
| | No | 176 98.9 | 62 35.2 | 44 25.0 | 42 23.9 | 23 13.1 | 5 2.8 | | 33 43.4 | 15 19.7 | 14 18.4 | 11 14.5 | 3 3.9 | |

We found a statistically significant association between age and GKOS at 1-year (*p* = 0.005), where 88.5% of geriatric patients had pain improvement following GKS, compared to 72.9% of adults. There was no significant association between gender and GKOS. Intriguingly, racial background was a statistically significant factor (*p* = 0.013) influencing outcome at 1-year, where people of Caucasian race demonstrated a better GKOS score. However, it is important to mention that most TN patients that underwent radiosurgery at our center were Caucasian (86%). In the same vein, the pain in the distribution of the various divisions of the trigeminal nerve was not associated with procedure outcome. Yet, patients experiencing atypical pain had a worse prognosis in the GKOS compared to those with typical pain, with the difference significant at 2 years. Although bilateral pain and atypical pain are associated with worse outcomes, the small sample makes accurate deductions difficult. In addition, we also checked the difference in outcome following GKS at 1-year and 2 years for patients with and without certain comorbidities (Table 4). Remarkably, a statistically significant association was seen between obesity and GKOS, where obese patients had worse outcomes compared to non-obese patients at both 1-year (*p* = 0.03) and 2 years (*p* = 0.005).

Pain recurrence rates were 16.9% at 1-year and 19.2% at 2 years. Facial numbness was present in 31.5% (56/178) and 30.8% (24/78) of patients at 1 and 2 years, respectively. In a similar manner, 37.6% (67/178) and 38.5% (30/78) of patients reported facial tingling at 12 and 24 months correspondingly.

## 4. Discussion

The International Association for the Study of Pain defines pain as "an unpleasant sensory and emotional experience associated with, or resembling that associated with, actual or potential tissue damage" [13]. Pain is a subjective experience influenced by biological, psychological and social factors. Due to the subjective nature of pain and differences in pain perception, we utilized a comprehensive scale to weigh the outcome of GKS by comparing a patient's pain level before and after the procedure. This was supplemented with evaluation of pain frequency and two pain severity scales, gauging the average and maximum pain experienced. The interpretation of GKS success varies from one study to another [9,14–16]. The distinguishing strength of the current study is the detail of follow-up pain assessment. A recent review study has shown that after GKS for TN, pain relief rates range from 70 to 98% with an average of 85.8%, over a mean or median follow-up period ranging from 17 to 68.9 months [9,15–33]. The beneficial effects of pain improvement after GKSRS diminishes with time. In the study by Sheehan et al. [9], some pain improvement after GKS was noticed in 90% of their patients at 1 year, which declined to 77% at 2 years and 70% at 3 years. This declining trend was also seen by Young et al. [26], where pain improvement was seen in 84.5%, 70.4% and 46.9% of their patients at 1, 3 and 5 years, respectively. In the same manner, in a long-term efficacy study, the actuarial probabilities of remaining pain free without medications were 71.8%, 64.9%, 59.7% and 45.3% at 3, 5, 7 and 10 years, respectively [28]. Since the success rate of GKSRS for treatment of medically intractable TN diminishes over time, we compared patients who were at the same follow-up stage. This helps in eliminating the waning effect of treatment when comparing patients for prognostic factors for treatment efficacy.

*Age:* In our series, older age (>60) was associated with a favorable radiosurgery outcome (*p* = 0.005) at 12 months. One year after radiosurgery, 88.5% of geriatric patients (>60) had a good outcome (grades 1A–C) compared to 72.9% of adults (18–60). Similarly, Sheehan et al. reported that increasing age correlated with a pain-free outcome [9]. Another study also supported these findings, showing that patient age older than 70 years was associated with a favorable response [28]. Finally, Karam et al. found a statistically significant correlation between age and recurrence of any pain, with age greater than 70 predicting a better outcome [34].

*Racial background:* Although in our cohort, patients of Caucasian descent demonstrated better outcomes at 1-year (grades 1A–C: 71.4% in African Americans vs. 85.7% in

Caucasians); most patients who underwent GKSRS and complied with follow-up were Caucasian (86% Caucasians, 11.8% African American and 4% Hispanic). A study by Reinard et al. [35], which retrospectively evaluated the medical records of patients, has shown that compared to Caucasians, African American patients were less likely to undergo MVD, percutaneous ablative procedures and SRS ($p < 0.001$). However, once seen by a neurosurgeon, there was no difference in likelihood of both patients to undergo a procedure. They concluded that racial disparities in management of TN appear to stem from a difference in referral patterns to neurologists and neurosurgeons. This could serve as an area of focus for future quality improvement initiatives to identify and reduce potential racial and socioeconomic disparities in management of TN [35].

*Obesity:* Obesity is associated with several pain disorders; it is documented to promote chronic pain, pain sensitization and neuropathic pain [36,37]. In murine models, obesity was shown to cause abnormal trigeminal sensory processing and nociceptive activation of the trigeminal system [38]. We found a statistically significant association between obesity and outcomes at 12 and 24 months. Although there were only a few patients with obesity in the present study, they had worse outcome grades as compared to non-obese patients. While underpowered to make definitive conclusions regarding the impact of obesity on outcomes, this finding suggests the need for additional research. Furthermore, a retrospective cohort analysis by Khattab et al. [36], demonstrated that elevated body mass index (BMI > 25) is associated with attenuated pain improvement following SRS for refractory TN. In the same manner, morbid obesity (BMI > 40) and diabetes were associated with higher reoperation rates following MVD for TN [39]. In the study by Marshall et al. [25], diabetes mellitus predicted decreased efficacy of GKS for TN, but we did not observe a significant association between diabetes and GKOS.

## 5. Conclusions

GKSRS is a safe and effective treatment option for patients with refractory TN. Our results demonstrate that most patients have a good response (grades 1A–C) at both 1-year (150 patients, 84.3%) and 2 years (64 patients, 82.1%) following radiosurgery. Geriatric age is associated with better radiosurgery outcomes, while obesity may be associated with a poorer prognosis.

**Author Contributions:** Conceptualization, A.J., K.D. and J.V.; methodology, A.J., K.D., J.B. and J.V.; software, A.J.; validation, A.J., R.C. and C.N.; formal analysis, A.J.; investigation, A.J., R.C. and C.N.; resources, R.C. and C.N.; data curation, R.C. and C.N.; writing—original draft preparation, A.J.; writing—review and editing, A.J., K.D., J.B. and J.V.; visualization, A.J.; supervision, K.D. and J.V.; project administration, A.J., K.D. and J.V. All authors have read and agreed to the published version of the manuscript.

**Funding:** This research received no external funding.

**Institutional Review Board Statement:** Ethical review and approval were waived for this study as we received a waiver of consent as a chart review from the institutional review board (IRB).

**Informed Consent Statement:** Patient consent was waived as we received a waiver of consent as a chart review from the institutional review board (IRB).

**Data Availability Statement:** Not applicable.

**Acknowledgments:** The authors wish to thank Anekay Kelly for the illustration included in this article.

**Conflicts of Interest:** The authors declare no conflict of interest.

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
