# Peer review of "Trigeminal Neuralgia Treatment Outcomes Following Gamma Knife Stereotactic Radiosurgery"

_2673-8937, doi:10.3390/ijtm2040041_

Round 1
Reviewer 1 Report
The authors presented their result of GKSRS for idiopathic trigeminal neuralgia by analyzing available 178 patients (30 percent of their treated patients ) from their prospecively collected database.
The authors have presented the information in a comprehensive manner, including the type of pain, exacerbating factors, and other detailed information that was in the database, which will be of some significance to the interested reader.
However, there does not appear to be any new information on the effectiveness of the GKSRS.
Although it is stated that obesity is involved in poor efficacy, the authors would do better to state the definition of obesity in Method.
In the Result, it is stated that patients received GKSRS treatment a mean of 7.8 years after the onset of trigeminal neuralgia, with a standard error of 0.58 years, which seems to be mistaken for a standard deviation.
Reviewer 2 Report
Dear Authors regarding your article Trigeminal Neuralgia Treatment Outcomes Following Gamma Knife Stereotactic Radiosurgery it is necessary to review some points.
Add the type of article in the title.
The introduction section is very short and is needed to add other references to increase the quality of the manuscript. I suggest some articles about orofacial pain and TMD that will be useful
Teledentistry in the Management of Patients with Dental and Temporomandibular Disorders Doi: https://doi.org/10.1155/2022/7091153
Prosthodontic Treatment in Patients with Temporomandibular Disorders and Orofacial Pain and/or Bruxism: A Review of the Literature https://doi.org/ 10.3390/prosthesis4020025
Efficacy of conservative approaches on pain relief in patients with temporomandibular joint disorders: a systematic review with network meta-analysis. PMID: 36148997.
Stem Cells in Temporomandibular Joint Engineering: State of Art and Future Persectives. The Journal of Craniofacial Surgery doi: 10.1097/SCS.0000000000008771
You need to review the grammar of the article
Add a table with the list of abbreviations
Kind Regards
